# Genetic and clinical variables act synergistically to impact neurodevelopmental outcomes in children with single ventricle heart disease

Thomas A. Miller[1,19]✉, Edgar J. Hernandez[2,19], J. William Gaynor[3], Mark W. Russell[4], Jane W. Newburger [5], Wendy Chung[6], Elizabeth Goldmuntz[7], James F. Cnota[8], Sinai C. Zyblewski[9], William T. Mahle[10], Victor Zak[11], Chitra Ravishankar[7], Jonathan R. Kaltman[12], Brian W. McCrindle[13], Shanelle Clarke[14], Jodie K. Votava-Smith[15], Eric M. Graham[9], Mike Seed [13], Nancy Rudd[16], Daniel Bernstein [17], Teresa M. Lee[6], Mark Yandell [2]✉ & Martin Tristani-Firouzi [18]✉

## Abstract

**Background** Recent large-scale sequencing efforts have shed light on the genetic contribution to the etiology of congenital heart defects (CHD); however, the relative impact of genetics on clinical outcomes remains less understood. Outcomes analyses using genetics are complicated by the intrinsic severity of the CHD lesion and interactions with conditionally dependent clinical variables.

**Methods** Bayesian Networks were applied to describe the intertwined relationships between clinical variables, demography, and genetics in a cohort of children with single ventricle CHD.

**Results** As isolated variables, a damaging genetic variant in a gene related to abnormal heart morphology and prolonged ventilator support following stage I palliative surgery increase the probability of having a low Mental Developmental Index (MDI) score at 14 months of age by 1.9- and 5.8-fold, respectively. However, in combination, these variables act synergistically to further increase the probability of a low MDI score by 10-fold. The absence of a damaging variant in a known syndromic CHD gene and a shorter post-operative ventilator support increase the probability of a normal MDI score 1.7- and 2.4-fold, respectively, but in combination increase the probability of a good outcome by 59-fold.

**Conclusions** Our analyses suggest a modest genetic contribution to neurodevelopmental outcomes as isolated variables, similar to known clinical predictors. By contrast, genetic, demographic, and clinical variables interact synergistically to markedly impact clinical outcomes. These findings underscore the importance of capturing and quantifying the impact of damaging genomic variants in the context of multiple, conditionally dependent variables, such as pre- and post-operative factors, and demography.

## Plain Language Summary

Single ventricle congenital heart disease is a birth defect. In these children, the heart has only one effective blood-pumping chamber instead of two. Surgery can reroute the blood to use only one chamber, but multiple risk factors influence how well a child develops afterwards. Studying these risk factors can be challenging because they are interconnected, i.e. children with a genetic birth defect may be more likely to have a lower birthweight, and hence more likely to spend longer in hospital after surgery. Here, we used a statistical approach not commonly applied to study congenital heart disease and describe that whether a genetic variant (a small difference in a child's DNA) is important for how a child with single ventricle heart disease develops and grows after surgery depends on the presence of other risk factors.

A full list of author affiliations appears at the end of the paper.

While survival for infants with single ventricle congenital heart disease (SV-CHD) has improved dramatically, significant healthcare utilization, morbidity, and mortality persist[1]. Despite improvements in standardized care, neurodevelopmental and growth outcomes vary substantially in children with SV-CHD[2]. Identifiable syndromes and chromosomal abnormalities are associated with worse outcomes for multiple forms of complex CHD[3–6]. Predicted damaging genetic variants in genes related to CHD may contribute to poor outcomes, irrespective of whether the patient has syndromic CHD. Outside of aneuploidy and other known syndromes, the genetic architecture of CHD includes high-impact, rare variants across many loci. Thus, many variants that contribute to CHD have a small population-attributable risk (PAR)[7, 8]. This low PAR makes determination of the association of variants with outcomes difficult, especially when studying relatively small numbers of individuals with SV-CHD. As a result, the relative contribution of genomic variation to neurodevelopment and growth outcomes remains poorly described, especially in the context of clinical variables and other comorbidities, which can combine with one another in a conditionally dependent manner to create a constellation of influence on any given outcome[9]. Two clinical variables are conditionally dependent if the probability of encountering both variables is not equal to the multiplication of their individual probabilities. For example, sex and bicuspid aortic valve (BAV) are conditionally-dependent, in that the probability of encountering an individual with bicuspid aortic valve is greater for males versus females[10].

We recently reported the conditionally dependent relationships between genotypes, gene functions, and phenotypes in a large CHD cohort using Bayesian Networks, an Artificial Intelligence (AI) solution[8]. Compared to traditional regression models, Bayesian Networks offer many advantages to risk prediction, as a single network can be used to evaluate multiple outcomes based upon any combination of variables or exposures[11,12]. Moreover, Bayesian Networks provide improved means to quantify the synergistic impacts of demographic, clinical, and genomic variables. In this context, we apply Bayesian Networks in an exploratory, proof-of-principle analysis to explore the landscape of damaging genotypes on neurodevelopmental and linear growth outcomes in the Pediatric Heart Network's (PHN) Single Ventricle Reconstruction (SVR) trial and Infant Single Ventricle (ISV) trial participants.

## Methods

**Study population.** Participants in the PHN's SVR and ISV trials who had blood or saliva collected for DNA extraction were included in this study. The original study design, descriptive characterization of the cohort, and primary results of the SVR (NCT00115934) and the ISV (NCT 00113087) trials have previously been published[13–17]. Both trials excluded participants with known chromosomal abnormalities or syndromic features at the time of enrollment. Institutional Review Boards for all centers participating in the SVR and ISV trials gave ethical approval for the human subjects research in this work and written informed consent was obtained. Data Use Agreements between the PHN and the Pediatric Cardiac Genomics Consortium (PCGC) allowed for the de-identified sharing of genetic and limited clinical data between the two consortia. All SVR and ISV participants who had whole exome sequencing (WES) and neurodevelopmental follow-up at 14 months of age were included in this cohort. Sub analyses were performed on those who had complete matching clinical and demographic variables.

**Clinical outcomes and covariates.** Outcomes included the Bayley Scales of Infant Development (BSID-II) and length measurement

(length for age $z$-score, LAZ) for both the SVR and ISV trials at 14 months of age. BSID-II subdomain scores included the Psychomotor Developmental Index (PDI) and Mental Developmental Index (MDI), wherein the standardized mean scores are 100. While it is true that MDI and PDI are assessed at the same time, we consider these variables within the Bayesian Networks as both outcomes and risk factors, in order to highlight the ability of our approach to capture and quantify synergistic (non-additive) relationships between variables under study. An advantage of our approach is that the use of a Bayesian network allows us to employ a single model to explore the impact of highly correlated variables singly or jointly, as outcomes and as risk factors to generate the results shown here. Additional pre- and post-operative clinical and demographic variables were available for a subset of 179 SVR participants, including birthweight, duration of mechanical ventilation following the Stage I palliation surgery, gestational age, and US census socioeconomic scores.

In order to optimally discretize continuous variables, we used a grid search to evaluate the performance of the nascent Bayesian Network, using receiver operating characteristic (ROC) curves (Supplementary Figure 1). For each value of MDI, PDI, and LAZ, we calculated the joint conditional probability of an individual having low MDI, PDI, or LAZ given the presence or absence of other conditions (variables) for each individual in the cohort given its intrinsic values at each condition. These conditional probability distributions were used to evaluate the accuracy of each candidate network to classify an individual as low or high MDI, PDI, or LAZ score, allowing us to select the optimal score resulting from the highest area under the ROC (AUC). This approach discovered that a score of 70 for both MDI and PDI resulted in an optimal AUC score (0.97 for PDI and 0.88 for MDI). A score of 70 corresponds to 2 standard deviations below the mean, consistent with clinical intuition for a desirable cut-off. The optimal value for LAZ score using this approach was −1.6 (AUC = 0.72). When considering a favorable outcome, we used a threshold of > 100 for MDI and PDI, and > 0 for LAZ. The pre- and post-operative clinical and demographic variables in the smaller subset of 179 SVR participants were less amenable to optimization using this approach. Therefore, we used previously published cutoffs for dichotomization previously associated with outcomes in this population[18,19] as follows: birthweight (<2500 gram, low), ventilation following the Stage I palliation surgery (>7 days, prolonged), gestational age (<37 weeks, preterm), and US census socioeconomic score (SES, low score <−0.3, Range −11.5 to 17.9).

**Variant annotation.** Raw sequencing data in the form of fastq files (Illumina 101 bp paired-end reads) for each individual were aligned to the human genome reference sequence (GRCh37/hg19, downloaded from http://genome.ucsc.edu) using the Burrows-Wheeler Alignment (BWA) Tool and the MEM algorithm (http://bio-bwa.sourceforge.net/bwa.shtml#13). Variants were called using the best practice protocol of the Genome Analysis Toolkit (GATK) pipeline (http://www.broadinstitute.org/gatk/guide/best-practices). All exomes were jointly-genotyped together with ~190 European individuals (CEU, GBR) from the 1000 Genomes Project. This was followed by Variant Quality Score Recalibration (VQSR), which uses known, high-quality variant sites from HapMap3 and 1000 Genomes to identify and filter potential false positive variant calls. Both variant calling and VQSR were performed separately on INDELs and single nucleotide variants (SNVs). Tranche values were set to 99.5 and 99.0 for SNPs and INDELs, respectively. Variant consequences and annotations were obtained with VEP v.95 (uswest.ensembl.org/info/docs/tools/vep/script/vep_download.html) utilizing ENSEMBL transcripts version 95

(excluding non-coding transcripts) and selecting the canonical transcript for analysis. Transcript-specific prediction for evaluating variant deleteriousness was calculated with VVP[20], which were also used as input for VAAST (VVP and VAAST: [github.com/Yandell-Lab/VVP-pub])[21]. Variants were annotated with Clin-Var (20200419) ensuring exact position and base match. Gene-conditions were extracted from the Online Mendelian Inheritance in Man (OMIM) catalog (2020_07)[22] and the Human Phenotype Ontology (HPO) database (obo file, 2020-08-11)[23]. Gene symbols were harmonized using ENSEMBL and HGNC databases controlling for synonymous gene symbols.

**AI-based scoring of damaging genetic variants**. AI-based prioritization and scoring of candidate disease genes and diagnostic conditions were performed using GEM[24] in the commercially available Fabric Enterprise platform (Fabric Genomics, Oakland, CA; fabricgenomics.com/fabric-gem). GEM inputs are genetic variant calls in VCF format and proband phenotypes in the form of HPO terms. GEM aggregates inputs from multiple variant prioritization algorithms (e.g., VAAST and VVP) with genomic and clinical database annotations, using Bayesian means to score and prioritize potentially damaged genes and candidate diseases. The prior probability for the model is based upon known disease associations in the Mendelian conditions databases OMIM and/or HPO with the gene in question. In order to capture a broad category of genes related to heart development, we selected the HPO term Abnormal Heart Morphology (HP:0001627).

GEM's gene scores are Bayes Factors[25], representing the $\log_{10}$ ratio between the posterior probabilities of two models, summarizing the relative support for the hypothesis that the prioritized genotype damages the gene in which it resides and explains the proband's phenotype versus the hypothesis that the variant neither damages the gene nor explains the proband's phenotype. We used a Bayes factor of ≥0.69 to represent a likely deleterious genetic variant, based on a recent retrospective analysis of critically ill newborns whereby a Bayes factor threshold ≥0.69 successfully achieved a diagnostic rate of 95% for cases solved using a standard clinical diagnostic approach[24]. Each variant that passed the 0.69 threshold was evaluated using the graph-based adjudication tool Graphite, which constructs a variant graph that permits the exclusion of false positives due to errors in variant calling (github.com/dillonl/graphite)[8]. Gene penetrance for GEM calculations was set to 0.95 to enforce strict consideration of known dominant and recessive disorders.

**Bayesian Network analyses**. Bayesian Network structure was learned using the R package "bnstruct"[26] which provides a Bayesian Information Criterion (BIC)-based exact structure search algorithm ([www.r-project.org] (bnlearn, gRain, bnstruct libraries, pROC package))[27]. The exact search algorithm explores the entire applicable space of conditional dependencies in order to discover the optimal network structure for the data. Parameter learning for this optimal network and multimorbidity risk calculations were accomplished using the loopy belief propagation algorithm from the R gRain package[28]. The conditional dependencies in the dataset are presented as a directed acyclic graph. Risk estimates derived from Bayesian Networks are maximum likelihood estimates given the optimal structure under the BIC and uniform prior probability. Relative risk was calculated in the following manner: Probability (A=True | B=True, C=True)/Probability (A=True | B=False, C=False). Confidence intervals (CI, ± 95%) for the mean values for these estimates were obtained by creating 1000 nets from bootstrap replicates of the same data.

*Statistical approach*. Differences in testing scores (MDI and PDI) and LAZ for those with and without damaging genetic variants in various gene categories were assessed using independent non-parametric Mann-Whitney U Test, correcting for multiple comparisons using a Bonferroni correction.

## Results

Of the original 555 SVR and 230 ISV trial enrollees, survival rates have been previously published (69% and 86%, respectively)[15,17]. Of those that underwent WES, there were 21 individuals who were enrolled in both SVR and ISV trials. For this study, a total of 304 participants underwent WES, neurodevelopmental assessment, and length measurement at 14 months of age and form the primary study cohort. Patient-parent trio WES was available for 86 participants. Additionally, the selected pre- and post-operative clinical and demographic variables were available for a subset of 179 SVR participants. Distribution of sex, race, ethnicity, cardiac anatomy, gestational age, and birthweight were similar across the various cohorts (Supplementary Table 1). The distribution of neurodevelopmental and growth scores for the cohort undergoing WES (Fig. 1) was similar to that previously reported for the original cohorts and underscore the highly affected nature of these patients[17,18].

A potentially damaging genetic variant in a gene related to "Abnormal Heart Morphology" was identified in 137 participants (44% of the study cohort), each harboring a mean of 1.8 damaging variants/proband (range 1–6; Supplementary Data 1). Mean MDI, PDI and LAZ scores were not statistically different between participants with and without a damaging genetic variant in this gene list (Fig. 2a). Forty-four subjects (14%) harbored one or more damaging genetic variants in a subset of the gene list that is associated with syndromic forms of CHD (as defined by OMIM), including autosomal dominant ($N=41$), X-linked recessive ($N=4$), X-linked dominant ($N=1$) and autosomal recessive ($N=1$) inheritance modes (Table 1). Eight of these subjects carried variants classified by ClinVar as pathogenic or likely pathogenic. The most commonly damaged genes in the cardiac syndromic subset gene list included *KMT2D* ($N=5$, Kabuki Syndrome), *NOTCH1* ($N=4$, Aortic Valve Disease 1), *ANKRD11* ($N=3$ KBG Syndrome), *MED12* ($N=2$, Opitz-Kaveggia Syndrome), *SHANK3* ($N=2$, Phelan-Mcdermid Syndrome), and *SRCAP* ($N=2$, Floating-Harbor Syndrome). While there was a trend toward lower MDI and PDI scores for participants with *versus* without a damaging variant in the syndromic CHD gene list, the differences were not significant after correction for multiple testing (Fig. 2b). We also examined damaging de novo genetic variants in the subset of participants with trio WES data ($N=86$). A damaging de novo genetic variant in the HPO-term derived gene list was identified in 10 probands, representing 12% of the cohort with trio WES data (Supplementary Table 2). The small number of de novo variants precluded statistical evaluation.

Next, we sought to investigate the conditional probability of particular outcomes, in the context of genetic, demographic and clinical variables, using Bayesian Networks. The Bayesian Network describing the landscape of genetic, neurodevelopmental, growth, and demographic variables in our cohort of SV-CHD (Fig. 3a) allows for a multitude of queries that support a robust investigation of multiple interactions between conditionally dependent variables. For example, querying the net reveals that a damaging variant in a gene related to abnormal heart morphology increased the risk of a low MDI score by 1.38-fold (1.37 CI-5%, 1.40 CI-95%, $N=1000$ bootstrap iterations; Fig. 3b and Supplementary Table 2). Of the variables under consideration, a low PDI score was the strongest single risk factor for a low MDI score (21.2-fold increase). While female sex was associated with a

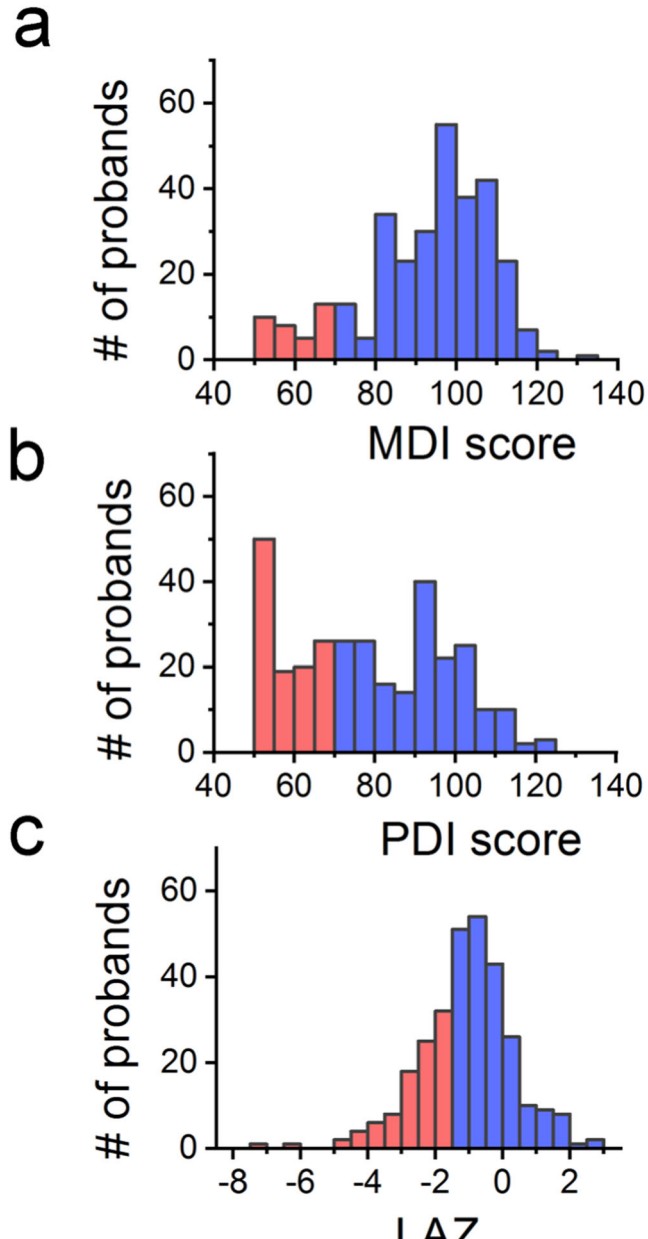

**Fig. 1 Outcomes Distribution for the study cohort.** Outcome distributions of the (**a**) Mental Developmental Index (MDI), (**b**) Psychomotor Developmental Index (PDI), and (**c**) length for age z-score (LAZ) for the study cohort. Red color denotes values below the optimally derived lower cut-off value (MDI, 70; PDI, 70; LAZ, −1.6) and blue those above the respective cut-off. See Methods for details. N = 309 subjects.

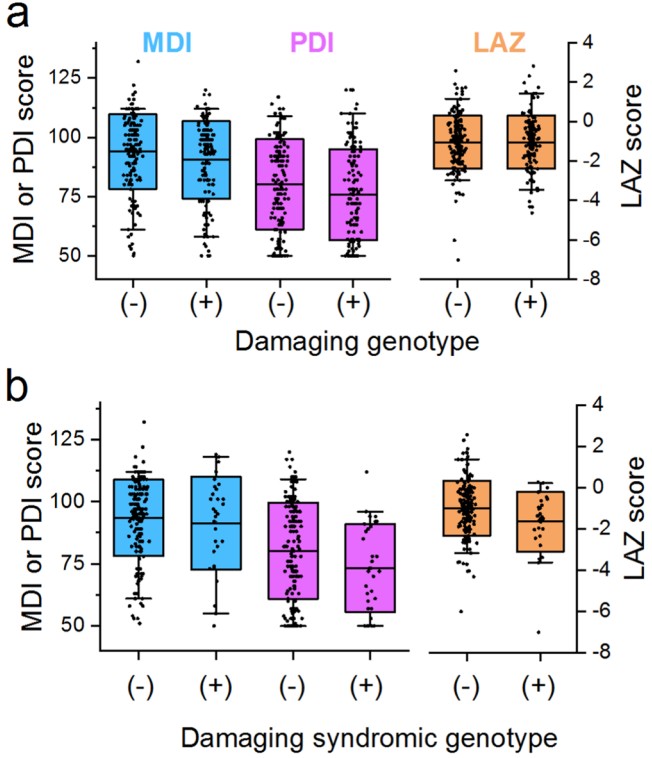

**Fig. 2 Neurodevelopmental and growth outcomes based on presence of a damaging genetic variant: distribution of Psychomotor Developmental Index (PDI), Mental Developmental Index (MDI), and length for age z-score (LAZ) for those individuals with and without a damaging genetic variant, defined by GEM Bayes Factor ≥0.69. a** Box plots of MDI, PDI and LAZ scores in participants with (+) and without (−) a damaging variant in a gene related to the HPO term "Abnormal Heart Morphology"; N = 167 participants without damaging genotype, N = 137 participants with damaging genotype. None of the comparisons achieved statistical significance after corrections for multiple testing: p = 0.94, MDI; p = 0.14, PDI; p = 0.94, LAZ. **b** Box plots of MDI, PDI and LAZ scores in participants with a damaging genotype in a known syndromic CHD gene (N = 38) and without (N = 167). Center horizontal line, mean value; box, 1 standard deviation; whiskers, 5 and 95th percentiles. None of the comparisons achieved statistical significance after corrections for multiple testing: p = 0.68, MDI; p = 0.15, PDI; p = 0.08, LAZ.

1.77-fold increased risk of a low MDI score in isolation, when combined with a low PDI score, the risk of a low MDI was further increased 75.8-fold. These results highlight the ability of Bayesian networks to quantify synergistic effects of variables in combination on an outcome of interest.

The contribution of genetics to PDI score was most impactful when considering predictors of a favorable PDI outcome (Supplementary Table 3 and Supplementary Fig. 2). Thus, a participant *without* a damaging variant in a syndromic CHD gene was 4-times more likely to have a normal PDI score (defined as >100). When combined with normal linear growth (LAZ > 0) and MDI ( > 100) scores, the absence of a damaging CHD syndromic genetic variant increased the probability of a normal PDI score by 16.5-fold.

Genetic information modestly impacted poor linear growth, with a damaging variant in a gene related to abnormal heart morphology or a syndromic CHD gene increasing the probability of a low LAZ score by 1.12- and 1.34-fold, respectively (Supplementary Table 3). Additionally, demographic and clinical variables interacted synergistically to increase the probability of a low LAZ score. For example, as isolated variables female sex and low MDI score increased the likelihood of a low LAZ score 1.19-fold and 2.13-fold, respectively. However, in combination, female sex and low MDI further increased the probability of a low LAZ score 6.4-fold.

Next, we sought to determine the impact of pre- and post-operative clinical variables, demographic variables such as SES and sex, and genetics both individually and in combination on neurodevelopmental and linear growth outcomes in a subset of 179 participants for whom complete variables were available. A network describing the landscape of clinical, genetic, demographic, length, and neurodevelopmental variable dependencies is depicted in Fig. 4. Data elements for this network include premature birth, birthweight, mechanical ventilation duration following the 1st stage palliation surgery and SES. Similar to the larger cohort, we were underpowered to detect the impact of a

**Table 1 Number of probands harboring a damaging variant in a syndromic CHD gene**

| Gene | # | MIM | Disease | Inheritance |
|------|---|-----|---------|-------------|
| KMT2D | 5 | 147920 | KABUKI SYNDROME 1 | AD |
| NOTCH1 | 4 | 616028 | AORTIC VALVE DISEASE 1 | AD |
| ANKRD11 | 3 | 148050 | KBG SYNDROME | AD |
| MED12 | 2 | 305450 | OPITZ-KAVEGGIA SYNDROME | XR |
| SHANK3 | 2 | 606232 | PHELAN-MCDERMID SYNDROME | AD |
| SRCAP | 2 | 136140 | FLOATING-HARBOR SYNDROME | AD |
| AFF4 | 1 | 616368 | CHOPS SYNDROME | AD |
| ARHGAP31 | 1 | 100300 | ADAMS-OLIVER SYNDROME 1 | AD |
| ARID1A | 1 | 614607 | COFFIN-SIRIS SYNDROME 2 | AD |
| ATN1 | 1 | 618494 | CONGENITA HYPOTONIA, EPILEPSY, DEVELOPMENTAL DELAY, AND DIGITAL ANOMALIES | AD |
| DVL3 | 1 | 616894 | ROBINOW SYNDROME, AUTOSOMAL DOMINANT 3 | AD |
| CDK13 | 1 | 617360 | CONGENITAL HEART DEFECTS, DYSMORPHIC FACIAL FEATURES, AND INTELLECTUAL DEVELOPMENTAL DISORDER | AD |
| CDK8 | 1 | 618748 | INTELLECTUAL DEVELOPMENTAL DISORDER WITH HYPOTONIA AND BEHAVIORAL ABNORMALITIES | AD |
| CHD7 | 1 | 214800 | CHARGE SYNDROME | AD |
| CREBBP | 1 | 180849 | BROAD THUMBS AND GREAT TOES, CHARACTERISTIC FACIES, AND MENTAL RETARDATION | AD |
| ECE1 | 1 | 613870 | HIRSCHSPRUNG DISEASE, CARDIAC DEFECTS, AND AUTONOMIC DYSFUNCTION | AD |
| EED | 1 | 617561 | COHEN-GIBSON SYNDROME | AD |
| EFTUD2 | | 610536 | MANDIBULOFACIAL DYSOSTOSIS, GUION-ALMEIDA TYPE | AD |
| ELN | 1 | 194050 | WILLIAMS-BEUREN SYNDROME | AD |
| EP300 | 1 | 613684 | RUBENSTEIN-TAYBI TYPE 2 | AD |
| FANCB | 1 | 314390 | VACTERL ASSOCIATION, X-LINKED, WITH OR WITHOUT HYDROCEPHALUS | XR |
| GLI3 | 1 | 146510 | PALLISTER-HALL SYNDROME | AD |
| LZTR1 | 1 | 616564 | NOONAN SYNDROME 10 | AD |
| MEIS2 | 1 | 600987 | CLEFT PALATE, CARDIAC DEFECTS, AND MENTAL RETARDATION | AD |
| MYRF | 1 | 618280 | CARDIAC-UROGENITAL SYNDROME | AD |
| NADSYN1 | 1 | 618845 | VERTEBRAL, CARDIAC, RENAL, AND LIMB DEFECTS SYNDROME 3 | AR |
| NF1 | 1 | 193520 | WATSON SYNDROME | AD |
| NSD1 | 1 | 117550 | SOTOS SYNDROME | AD |
| OFD1 | 1 | 311200 | OROFACIODIGITAL SYNDROME I | XD |
| PBX1 | 1 | 617641 | CONGENITAL ANOMALIES OF KIDNEY AND URINARY TRACT SYNDROME WITH OR WITHOUT HEARING LOSS, ABNORMAL EARS, OR DEVELOPMENTAL DELAY | AD |
| POGZ | 1 | 616364 | WHITE-SUTTON SYNDROME | AD |
| SMC1A | 1 | 300590 | CORNELIA DE LANGE SYNDROME 2 | XR |
| SOS1 | 1 | 610733 | NOONAN SYNDROME 4 | AD |
| TBX2 | 1 | 618223 | VERTEBRAL ANOMALIES AND VARIABLE ENDOCRINE AND T-CELL DYSFUNCTION | AD |
| TBX5 | 1 | 142900 | HOLT-ORAM SYNDROME | AD |

Damaged genes associated with syndromic CHD (as defined by OMIM) recovered in the study cohort are listed.
*MIM* phenotype number as listed in OMIM, *AD* autosomal dominant, *AR* autosomal recessive, *XR* X-linked recessive, *XD* X-linked dominant.

damaging variant in the syndromic CHD gene list on low MDI, given that only 5 probands harbored a damaging syndromic genetic variant and had an MDI score ≤70. As isolated variables, a damaging genetic variant in a gene related to abnormal heart morphology and prolonged ventilator support following stage I palliative surgery increased the probability of having a low MDI score by 1.92- and 5.76-fold, respectively. However, in combination, these variables acted synergistically to further increase the probability of a low MDI score by 10.5-fold. (Fig. 4 and Supplementary Table 4). Likewise, genetic and clinical variables interacted synergistically to predict a favorable neurodevelopmental outcome. For example, the absence of a damaging variant in a known syndromic CHD gene and shorter post-operative ventilator support increased the probability of a normal MDI score 1.74- and 2.38-fold, respectively, but in combination increased the probability of a good outcome by 59.4-fold.

Similar to the larger cohort, the contribution of genetics to PDI score was most impactful when considering predictors of a favorable PDI outcome (Supplementary Table 4 and Supplementary Fig. 3). A participant without a damaging variant in a syndromic CHD gene was 1.39-times more likely to have a normal PDI score. However, when combined with a high SES score, the absence of a damaging CHD syndromic genetic variant increased the probability of a normal PDI score by 52.8-fold.

The genetic signal for poor linear growth was smaller in this cohort, in part due to loss of power in the smaller dataset. The strongest single variable contributing to poor linear growth was a low a SES score, increasing the probability of a low LAZ score by 2.13-fold (Fig. 4 and Supplementary Table 4). When combined with female sex and a low PDI score, a low SES score increased the probability of a low LAZ score by 7.78-fold.

## Discussion

Exploring heterogeneity in outcomes following surgery for complex CHD necessitates models capable of quantifying the complex relationships between clinical variables and genetics, and their combined impacts on divergent outcomes. This is because clinical variables often combine with each other in a conditionally-dependent manner to impact the probability of any given outcome[9]. Here, we apply AI methodologies as a proof-of-principle study of the impact of genetics on neurodevelopmental and linear growth in SV-CHD, using AI-based prioritization of damaging genetic variants and a Bayesian Network approach. Our analyses suggest a modest genetic contribution to neurodevelopmental outcomes as isolated variables, similar to known clinical predictors. By contrast, we discovered that genetic, demographic, and clinical variables interact synergistically to

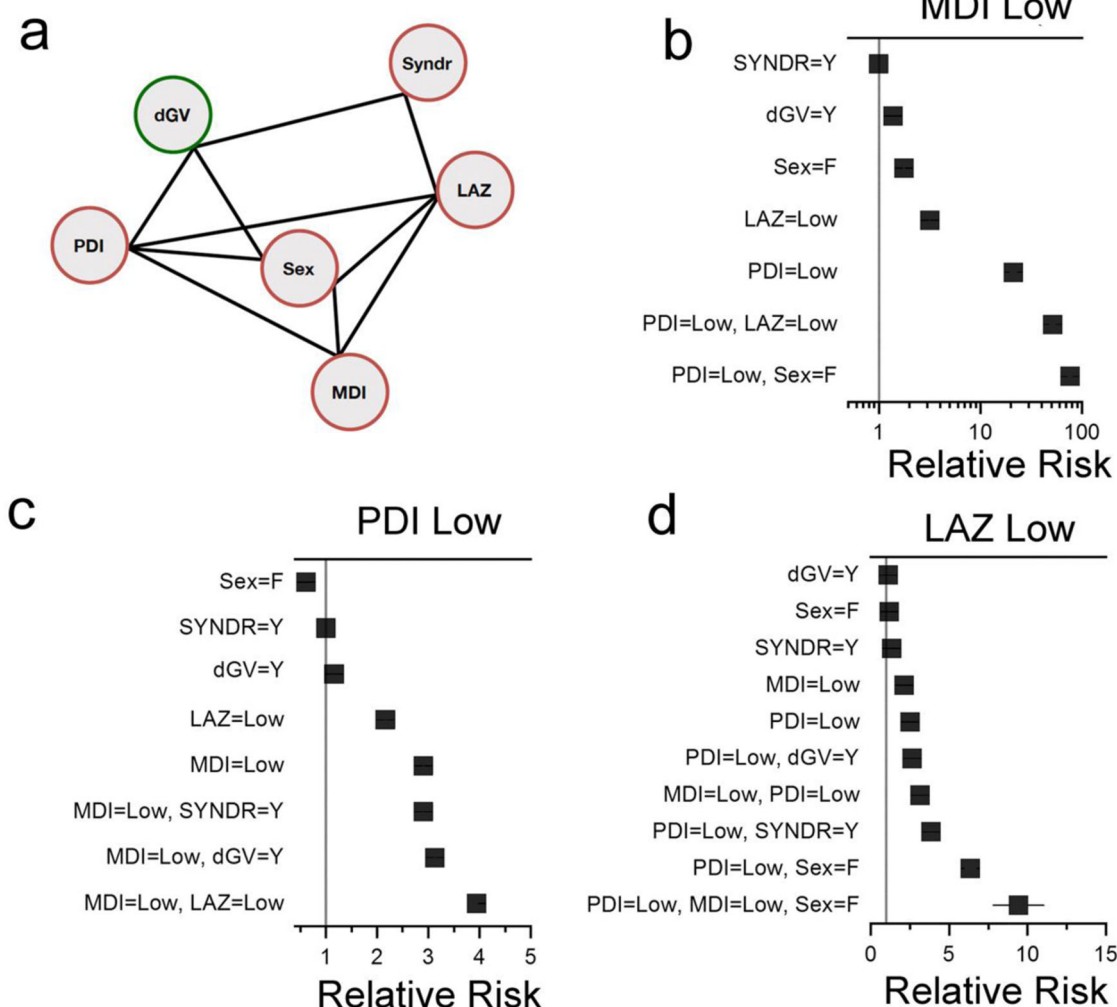

**Fig. 3 Network describing the conditional dependencies among the genetic, demographic and selected clinical variables in a cohort of children with single ventricle CHD. a** Best-fit network for the variables under study. Edges between nodes indicate conditional dependencies. dGV, damaging genetic variant in genes associated with HPO term "Abnormal Heart Morphology". SYNDR, damaging genetic variant in genes associated with syndromic CHD, defined by OMIM. LAZ, length for age Z-score. $N = 309$ participants. **b–d** Forest plots showing relative risk ratios for low MDI ($\leq$70), PDI ($\leq$70), and LAZ ($\leq$−1.6) scores in the context of the displayed clinical variables alone or in combination. Solid black line denotes 5 and 95% confidence intervals. If not visible, confidence intervals are within the symbol.

markedly impact clinical outcomes. These findings highlight the importance of evaluating genetic data in the context of clinical variables, and the need to employ approaches to capture synergistic relationships between them.

Our analyses reveal a modest genetic contribution to poor neurodevelopmental outcomes in probands harboring a damaging variant in abnormal heart morphology genes versus syndromic CHD genes. In the context of predicting a favorable outcome, absence of a damaging variant in the syndromic CHD gene list was more impactful. Although individuals with known syndromic disorders were excluded from enrollment in the SVR and ISV trials, we found that 14% of the SV-CHD cohort harbored a damaging genetic variant in a gene known to cause syndromic CHD, including *KMT2D*, *NOTCH1,* and *ANKRD11* as the most common recurring damaged genes. Similar to this cohort, the PCGC also reported a higher-than-expected number of participants with damaging genotypes in syndromic genes, despite a known genetic syndrome as an exclusion criterion[7]. The inherent variable expressivity of cardiac syndromic disorders and/ or potential challenges in identifying syndromic features in infancy may, in part, explain these findings. Importantly, our

estimates of the impacts of damaging genotypes in syndromic genes (or their absence) are underestimated by the initial exclusion of known syndromic patients. The power of this Bayesian approach is that we can tease apart the contribution of syndromic genes (or other gene classes) to clinical outcomes and thereby predict risk across all patients.

An advantage of Bayesian Networks is the ability to capture and quantify the joint contributions of multiple conditionally dependent variables, all within a single probabilistic graphical model. Traditional techniques, such as regression analyses, have limited ability to capture the conditional dependencies between clinical variables. While mixture and generalized linear models can overcome these weaknesses, they require the design of a new model to address each individual question. Neural nets, while capable of capturing conditional dependencies, lack explainability in that the contributions of different variables on a given output are often unclear. By contrast, Bayesian Networks provide an explainable AI solution[29].

Our Bayesian Network approach allowed us to start to describe the landscape of clinical, genetic and demographic variables in children with SV-CHD, capturing and quantifying the impacts of

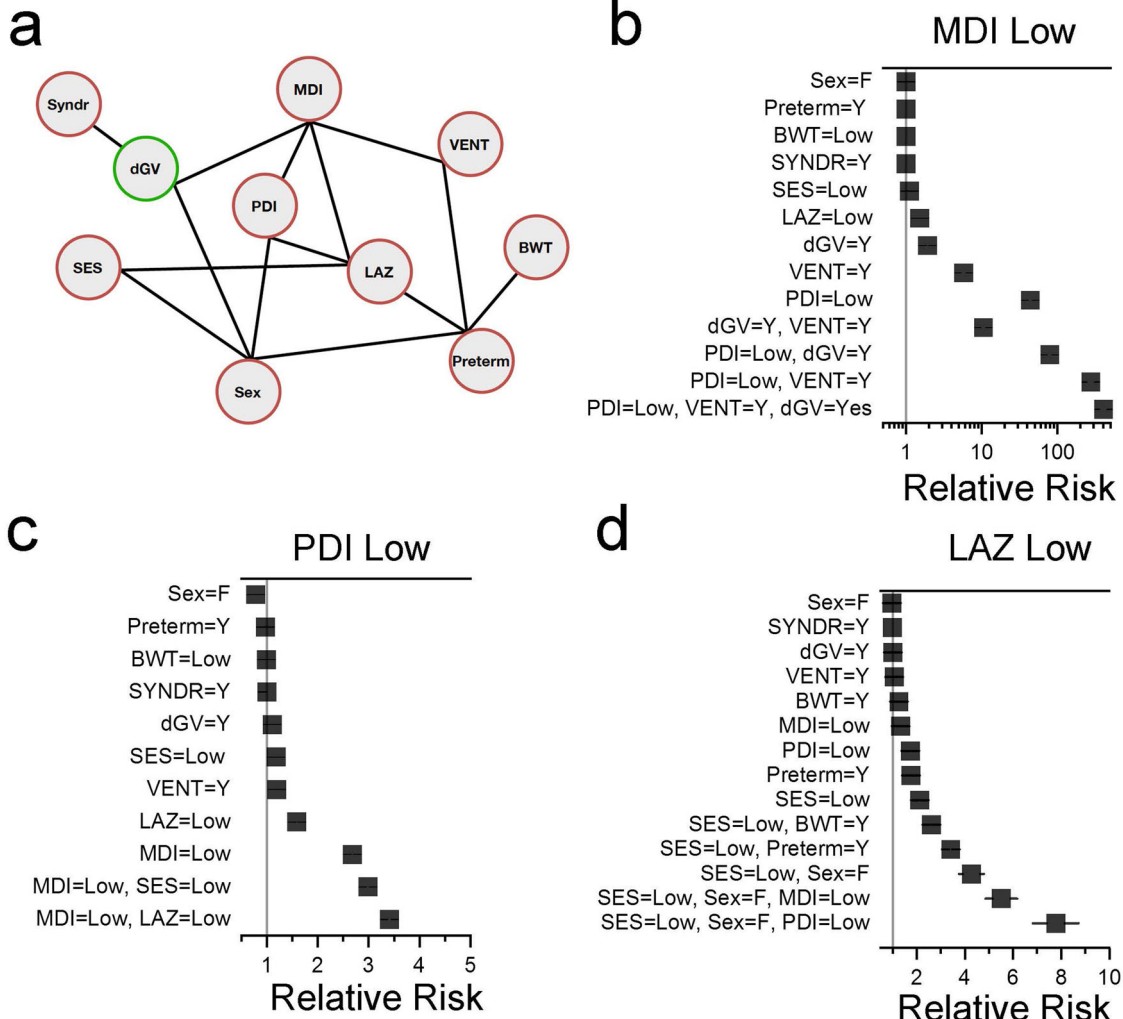

**Fig. 4 Network describing the conditional dependencies among pre- and post-operative clinical variables, genetics and demographic features in children with single ventricle CHD. a** Best-fit network for the variables under study. Edges between nodes indicate conditional dependencies. BWT birth weight (<2500 grams, low BWT), VENT mechanical ventilator duration after 1st stage single ventricle palliation (>7 days, prolonged VENT), preterm preterm birth (<37 weeks gestation), SES socioeconomic status score; $N = 179$ participants. **b**–**d** Forest plots showing relative risk ratios for low MDI (≤70), PDI (≤70), and LAZ (≤−1.6) scores in the context of the displayed clinical variables alone or in combination. Solid black line denotes 5 and 95% confidence intervals. If not visible, confidence intervals are within the symbol.

synergistic relationships between these variables and outcomes of interest. Genetic variables interacted synergistically with duration of post-operative mechanical ventilation to increase the probability of a poor or favorable MDI score by 10- and 59-fold, respectively. Likewise, genetic variables interacted synergistically with SES to increase the likelihood of a favorable PDI score by 53-fold. Our analyses here were limited to the relatively small number of individuals with complete clinical variables; precluding further exploration of the impact of a damaging genetic variant on pre- and post-operative variables, as well as SES. Expanding the analysis to the wider range of clinical variables associated with this well-phenotyped cohort and increasing the number of genotyped probands will provide greater insight into the genetic contribution to clinical outcomes in SV-CHD. In a broader sense, the current work supplements our efforts using Bayesian Networks as an explainable AI solution for understanding the joint impacts of diagnoses, medications, and medical procedures on cardiovascular health outcomes[9].

There are several limitations inherent to this study. For example, DNA collection was not a requirement for participation and thus, sample procurement was inconsistent across the cohort.

Moreover, DNA samples were not obtained at enrollment, but rather at subsequent time points in these longitudinally followed cohorts, thereby introducing survival bias. Some DNA samples were collected at a few years of age which would have been well after the 14-month clinical endpoints included in these analyses. Thus, patients who died early may have harbored highly damaging genotypes that are not accounted for in this cohort of infants that survived at least until the 14-month endpoint. The later DNA analysis also highlights the limitations of the original exclusion criteria, as some individuals in SVR were later found (during enrollment in SVR extension studies) to have syndromes, such as Kabuki Syndrome, consistent with our variant analysis. Additionally, we recognize the relatively small sample size of patients limited the ability to include all potential peri-operative, clinical, and socio-economic predictors of outcomes in this proof-of-principal analysis. Future efforts by the PHN consortium to sequence more patients with rich clinical datasets as well as by the PCGC to obtain richer clinical data on their sequenced patients, will allow for more robust analyses of the genetic contribution to clinical outcomes in children with complex forms of CHD. The definition of a "damaging" genetic variant is a topic of intense

debate[30]. Our rationale for defining a damaging variant was based on a recent study benchmarking the AI tool against clinically solved cases[24]. Stricter criteria for a damaging classification will allow for fewer false positives, but at the expense of false negatives and *vice versa*. However, the expectation is that false positives will dilute genetic signals, but truly benign variants are a priori not positively correlated with negative outcomes. Thus, the magnitudes and statistical significance of the associations between damaging genotypes and clinical variables reported here should be thought of as lower bounds. Finally, the goal of this study was to highlight the advantages of an AI-based Bayesian Networks approach as a proof-of-principle analysis, leveraging existing ISV and SVR clinical and genetic data. In this context, the risk predictions are directly relevant to the study cohort, but cannot be more widely applied without a validation cohort.

Collectively, our findings highlight how genetics, demography, and clinical variables interact synergistically to impact neurodevelopmental outcomes and underscore the importance of capturing and quantifying these synergies. We have also demonstrated the utility of Bayesian networks for conditional risk analyses. Moving forward, large-scale collaborations between genetic and clinical consortia will provide more extensive datasets to better understand the synergistic impacts of genetic and clinical variables on clinical outcomes in children with CHD.

**Reporting summary**. Further information on research design is available in the Nature Portfolio Reporting Summary linked to this article.

## Data availability

The sequencing data used in this analysis may be downloaded, with committee-approved access, from the database of Genotypes and Phenotypes (dbGaP) [www.ncbi.nlm.nih.gov/] (accession numbers phs000571.v5.p2). Clinical data for this cohort is available as a public use dataset (https://www.pediatricheartnetwork.org/login/). Source data for the main figures are available as Supplementary Data.

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

## Acknowledgements

The clinical data for this project was supported by National Heart, Lung, and Blood Institute (NHLBI) Pediatric Heart Network grants HL068269, HL068270, HL068279, HL068281, HL068285, HL068288, HL068290, HL068292, and HL085057. The genomic data for this project was supported by the NHLBI Pediatric Cardiac Genomics Consortium (UM1-HL098147, UM1-HL128761, UM1-HL098123, UM1-HL128711, UM1-HL098162, U01-HL131003, U01-HL098153, U01-HL098163), the National Center for Research Resources (U01-HL098153), and the National Institutes for Health (R01-GM104390, 1S10OD021644-01A1). The content of this article is the responsibility of the authors and do not represent the official views of the National Heart, Lung, and Blood Institute or the NIH.

## Author contributions

T.A.M., E.J.H., J.W.G., M.W.R., M.Y., and M.T.F. conceived and designed the study. T.A.M., E.J.H., M.Y., and M.T.F. performed the bulk of the analysis and drafted the initial manuscript. J.W.G., M.W.R., J.W.N., W.C., E.G., J.F.C., S.C.Z., W.T.M., V.Z., C.R., J.R.K., B.W.M., S.C., J.K.V-S., E.M.G., M.S., N.R., D.B., and T.M.L. provided ongoing review of and feedback on analyses over the course of multiple meetings, and reviewed and revised the manuscript.

## Competing interests

M.Y. is a consultant to Fabric Genomics Inc. and has received consulting fees and stock grants from Fabric Genomics Inc. The remaining authors declare that they have no competing interests relevant to this project.

## Additional information

[1]Department of Pediatrics, Maine Medical Center, Portland, ME, USA. [2]Department of Human Genetics and Utah Center for Genetic Discovery, University of Utah, Salt Lake City, UT, USA. [3]Department of Surgery, Children's Hospital of Philadelphia, and the Perelman School of Medicine, University of Pennsylvania, Philadelphia, PA, USA. [4]Department of Pediatrics, University of Michigan, Ann Arbor, MI, USA. [5]Department of Cardiology, Boston Children's Hospital, Department of Pediatrics, Harvard Medical School, Boston, MA, USA. [6]Departments of Pediatrics and Medicine, Columbia University, New York, NY, USA. [7]Division of Cardiology, Children's Hospital of Philadelphia, Department of Pediatrics, Perelman School of Medicine, University of Pennsylvania, Philadelphia, PA, USA. [8]Heart Institute, Cincinnati Children's Hospital, Cincinnati, OH, USA. [9]Department of Pediatrics, Medical University of South Carolina, Charleston, SC, USA. [10]Children's Healthcare of Atlanta, Atlanta, GA, USA. [11]Healthcore, Watertown, MA, USA. [12]Division of Cardiology, Children's National Hospital, Washington, DC, USA. [13]Labatt Family Heart Centre, The Hospital for Sick Children, Department of Pediatrics, University of Toronto, Toronto, ON, Canada. [14]Department of Pediatrics Emory University School of Medicine, Atlanta, GA, USA. [15]Department of Pediatrics, Children's Hospital Los Angeles, Los Angeles, CA, USA. [16]Department of Pediatrics, Medical College of Wisconsin, Milwaukee, WI, USA. [17]Stanford University School of Medicine, Stanford, CA, USA. [18]Department of Pediatrics, University of Utah, Salt Lake City, UT, USA. [19]These authors contributed equally: Thomas A. Miller, Edgar J. Hernandez. ✉email: Thomas.A.Miller@mainehealth.org; myandell@genetics.utah.edu; Martin.Tristani@utah.edu

