## [Peer Review File · Communications Medicine]

Genetic, demographic and clinical variables act synergistically to impact neurodevelopmental outcomes in children with single ventricle heart diseaseReviewers' comments:

Reviewer #1 (Remarks to the Author):

This is an interesting manuscript using sophisticated AI methods to investigate the conditional dependencies among genetic, demographic and clinical variables to predict neurodevelopmental outcomes in a cohort of 14-month-old children with univentricular CHD. By using Bayesian networks, the authors could show that genetic vulnerability interacts with demographic and clinical variables and impacts neurodevelopmental outcome. Importantly, this paper does not only investigate neurodevelopmental impairments but also favorable outcomes.

While the authors are to congratulate to their novel approach, one main concern relates to the fact that the authors used a data-driven approach and did not validate their model in a separate dataset. Further, the authors do not justify the selection of risk factors that were included in their analysis. Therefore, the generalizability of their findings is debatable. Additional questions and suggestions stated below may improve the manuscript's clarity:

Abstract

The abstract does not contain any information on the cohort size and any other demographic information. Further, there are no CI for the increase in probabilities of adverse outcome. No information is given how "low" MDI was defined and that the MDI is part of the Bayley Scales of Infant Development II. Also, a brief information on the genetic analyses performed in this sample should be provided in the Abstract.

Introduction

1. Please state in the aims (end of introduction section) that this analysis is of exploratory nature as you do not provide any specific hypotheses and your statistical approach is data-driven.

Methods

2. I suggest to describe the recruitment procedure in a flow chart especially since you analyze different subsets of patients. Visualizing the number of patients who underwent the 14-month follow up, got genetic sequencing, triplets, and clinical data would be informative and easier to follow. Also, please compare the participating subjects to those who dropped out in order to make assumptions about a potential ascertainment bias (e.g., SES, clinical characteristics).

3. Birthweight and gestational age were considered as a clinical variable. As these variables correlate strongly, was birthweight adjusted for gestational age – that is expressed as z-scores?
4. How were the clinical variables selected and why was length of hospitalization not included considering that length of hospitalization is one of the most established clinical predictors for neurodevelopmental outcomes in CHD? Did the authors examine collinearity of clinical variables?
5. How was “damaging gene variant” defined? Was this defined by the geneticists after obtaining clinical data and were some of the patients excluded from the analysis once their genetic comorbidity was diagnosed? See patients listed in Table 1, with e.g. Kabuki syndrome, Opitz syndrome, ChARGE syndrome? As stated in the methods section, the original study excluded patients with a genetic syndrome. Accordingly, patients with a newly detected genetic syndrome should be excluded or at least the inconsistency should be discussed in the limitation section.

Results

6. Please comment on the 14-month survival rates of the two trials. Are these rates comparable to other studies and is there an explanation for the difference between the two cohorts (57% and 73%)? This may inform the reader about the generalizability of the study results.
7. The authors state that “Of the 555 and 230 individuals enrolled in the SVR and ISV trials, respectively, 57% and 73% survived to the 14-month endpoint and 36% and 67% underwent WES”. Does the 36% and 67% refer to the 555 and 230 individuals? This is not clear.
8. Can the authors provide any information why exome sequencing was not conducted in a large proportion of patients?
9. In line 223 the authors state that the subset of patients with trio exome sequencing is n = 108. In line 246 it is stated that the subset of patients with trio exome sequencing is n = 86. Why are 22 patient trios missing in this analysis?
10. In the network analysis, MDI and PDI are included and interpreted as outcomes and risk factors. Since MDI and PDI are assessed at the same time, the authors cannot infer about a causal pathway and therefore these variables should not be interpreted as “risk factors” for the other outcome. I suggest to either calculate separate networks for each outcome measure (PDI, MDI, LAZ) or not using the term “risk factor” for outcome variables.
11. Since the authors use a data-driven machine learning /AI approach to investigate the conditional dependencies among genetic, demographic and clinical variables, the model should be validated in an independent dataset. Could the authors discuss this in the discussion section.
12. Please provide confidence intervals in Figure 3 and 4 as well as when you report relative risks in the text.

Discussion

13. Please comment on female sex being a significant risk factor for poorer neurodevelopmental outcome in the model. This is unusual as most previous studies identified male sex a risk factor for poorer neurodevelopmental outcome.

Reviewer #2 (Remarks to the Author):

This study uses Artificial Intelligence methods to describe the intertwined relationships between clinical variables, demography, and genetics in a cohort of children with single ventricle CHD. Although addressing the impact of genetics on clinical outcomes and especially neurodevelopmental outcomes is very important as it remains poorly understood, this study suffers from limitations in achieving this.

The use of Bayesian Networks allows to capture and quantify the joint probability of multiple conditionally dependent variables within the same model. However, it requires an accurate classification of all variables. The authors do not provide detailed information on the discretisation of many of the continuous variables.

In terms of the genetics, the model explored the presence of damaging genetic variant or not within each sample in the cohort. The use of AI-methodology resulted in finding a potentially damaging variant in 44% of the cohort. This seems very high- there has not been any genetic study in CHD (but also in most complex diseases) that has been able to identify such a large percentage. I think this approach includes variants (especially missense) variants that are most probably benign.

The authors comment on the CHD cohort consisting of mainly non-syndromic CHD cases. However, they search for genetic variants in syndromic CHD genes. There is no exploration of matching of phenotypes between the cases and the genes. In Table 1 (and Sup Table 1), the diseases associated with the variants found are very specific – have there been any exploration of any phenotypic features in the corresponding cases that harbour these variants?

A quick exploration of these variants showed that several of them are not predicted pathogenic by multiple pathogenicity prediction algorithms (I used varsome). So, I argue that this approach includes variants that might not be contributing at all the to cases' phenotype.

In my opinion, a stricter variant prioritisation is required. However, this would result in a much lower percentage of cases with potentially pathogenic variants and given the size of the cohort, might make the current analysis impossible.

RESPONSE TO REVIEWERS

We thank the Reviewers and Editors for their thoughtful and insightful comments and have incorporated their comments into the body of the manuscript. A summary of the reviewer comment and our point-by-point response are listed below.

Reviewer #1:

While the authors are to congratulate to their novel approach, one main concern relates to the fact that the authors used a data-driven approach and did not validate their model in a separate dataset. Further, the authors do not justify the selection of risk factors that were included in their analysis. Therefore, the generalizability of their findings is debatable. Additional questions and suggestions stated below may improve the manuscript's clarity:

Thank you for this comment and support. Please see additional specific responses below, but we agree that the findings for this specific cohort cannot be generalized without a validation cohort. We have made additions to the text as suggested in specific comments below to clarify the proof-of-principle conclusions and lack of generalizability including in the limitations paragraph that now reads:

“Finally, the goal of this study was to highlight the advantages of an AI-based Bayesian Networks approach as a proof-of-principle analysis, leveraging existing ISV and SVR clinical and genetic data. In this context, the risk predictions are directly relevant to the study cohort, but cannot be more widely applied without a validation cohort.”

Abstract

The abstract does not contain any information on the cohort size and any other demographic information. Further, there are no CI for the increase in probabilities of adverse outcome. No information is given how “low” MDI was defined and that the MDI is part of the Bayley Scales of Infant Development II. Also, a brief information on the genetic analyses performed in this sample should be provided in the Abstract.

While we agree these are important details for an abstract, the word count limitations prevent us from including additional details. We will defer to the Editors for guidance.

Introduction

Please state in the aims (end of introduction section) that this analysis is of exploratory nature as you do not provide any specific hypotheses and your statistical approach is data-driven.

The last sentence of the introduction has been edited to now read:

“In this context, we apply Bayesian Networks in an exploratory, proof-of-principle analysis to explore the landscape of damaging genotypes on neurodevelopmental and linear growth outcomes in the Pediatric Heart Network’s (PHN) Single Ventricle Reconstruction (SVR) trial and Infant Single Ventricle (ISV) trial participants.”

The first paragraph of the Discussion now reads as follows:

“Here, we apply AI methodologies **as a proof-of-principle study** of the impact of genetics on neurodevelopmental and linear growth in SV-CHD, using AI-based prioritization of damaging genetic variants and a Bayesian Network approach”.

Methods

I suggest to describe the recruitment procedure in a flow chart especially since you analyze different subsets of patients. Visualizing the number of patients who underwent the 14-month follow up, got genetic sequencing, triplets, and clinical data would be informative and easier to follow. Also, please compare the participating subjects to those who dropped out in order to make assumptions about a potential ascertainment bias (e.g., SES, clinical characteristics).

Based on this concern combined with additional concerns raised in subsequent comments below, we have added Supplementary Table 1 that compares basic demographics from the original cohorts enrolled in both trials, the larger analytic cohort, the cohort with the complete clinical covariates, and the cohort with trio data. Supplementary Table 1 contains the following information:

		SVR and ISV Trial Enrollment Cohorts (n= 785)	Analytic Cohort with MDI/PDI/Length (n=304)	Cohort with full covariates (n=179)	Trios (n=81)
Female Sex (%)		36	35	38	35
Race (%)	White	80	84	88	89
	Black	15	12	9	5
	Asian	2	1	1	2
	Other	3	4	2	4
Ethnicity (%)	Hispanic	17	15	14	12
	Non - hispanic	81	84	86	88
	Other	2	1	0	0
HLHS (%)	Yes	81	76	88	85
	No	19	24	12	15
Mean Gestational Age (weeks)		38.0	38.3	38.3	38.2
Mean Birthweight (grams)		3149	3187	3097	3076

Birthweight and gestational age were considered as a clinical variable. As these variables correlate strongly, was birthweight adjusted for gestational age – that is expressed as z-scores?

For this analysis we only used dichotomized variables not adjusted for gestational age. We agree that birthweight and gestational age are highly correlated, but have previously published in the SVR cohort that categorical cut-offs for low birthweight and prematurity, as well as size for gestational age are uniquely associated with differences in various clinical outcomes (Miller, et al, Pediatrics 2019). This reference has been added to the Methods section for justification of the variable selection which now reads:

“We used previously published cutoffs for dichotomization previously associated with outcomes in this population^{18,19} as follows: birthweight (< 2500 gram, low), ventilation following the Stage I palliation surgery (> 7 days, prolonged), gestational age (<37 weeks, preterm), and US census socioeconomic score (SES, low score < -0.3, Range -11.5 to 17.9).”

How were the clinical variables selected and why was length of hospitalization not included considering that length of hospitalization is one of the most established clinical predictors for neurodevelopmental outcomes in CHD?

To some degree, we were limited in our ability to obtain the complete set of clinical variables from these cohorts, based on the data agreements signed by the Pediatric Cardiac Genomics Consortium (PCGC) and the Pediatric Heart Network (PHN) at the time of this project’s initiation. Given the close correlation between ventilator days, ICU LOS and hospital LOS, we elected to use ventilator days. The lack of LOS variables, along with other potential important variables is now commented on in the Limitations section:

“Additionally, we recognize the relatively small sample size of patients limited the ability to include all potential peri-operative, clinical, and socio-economic predictors of outcomes in this proof-of-principal analysis. Future efforts by the PHN consortium to sequence more patients with rich clinical datasets as well by the PCGC to obtain richer clinical data on their sequenced patients, will allow for more robust analyses of the genetic contribution to clinical outcomes in children with complex forms of CHD....”

Did the authors examine collinearity of clinical variables?

We are aware of highly associated clinical variables in the dataset and while collinearity is an undesired feature in regression models, it does not impede analyses using Bayesian Networks. Collinearity directly impacts the interpretation of regression models by obscuring the individual effects of a predictor variable on the response variable. In the case of Bayesian Networks, the joint probability distribution of the entire network is evaluated, providing the means to explore the conditional dependency between pairs of variables (or multivariable queries) without influence from their correlation with other variables and quantify the impacts single or multiple variables on a target variable.

How was “damaging gene variant” defined?

As stated in the Methods section, “We used a Bayes factor of ≥ 0.69 to represent a likely deleterious genetic variant, based on a recent retrospective analysis of critically ill newborns whereby a Bayes factor threshold ≥ 0.69 successfully achieved a diagnostic rate of 95% for cases solved using a standard clinical diagnostic approach (De La Vega, F.M. *et al.* Artificial intelligence enables comprehensive genome interpretation and nomination of candidate diagnoses for rare genetic diseases. *Genome Medicine* **13**, 153, 2021). **See detailed response to Reviewer 2 below for more details on changes made to the manuscript to clarify the approach to defining a variant as damaging and the limitations to various approaches.**

The original study excluded patients with a genetic syndrome. Accordingly, patients with a newly detected genetic syndrome should be excluded or at least the inconsistency should be discussed in the limitation section.

Indeed, the original SVR and ISV studies excluded patients with a known genetic syndrome. The observation that we detected damaging variants in known syndromic genes highlights the subtle nature of phenotypic manifestations of syndromic disorders in newborns/infants. Likewise, the Pediatric Cardiac Genomics Consortium (PCGC) excluded patients with known syndromic disorders and yet recovered a number of damaging syndromic genotypes in their cohort. Importantly, our estimates of the impacts of damaging genotypes in syndromic genes (or their absence) are underestimated by the initial exclusion criterion. The strength of the Bayesian approach is that we can tease apart the contribution of syndromic genes (or other gene classes) to clinical outcomes and thereby predict risk across all patients. We now include these points in the Discussion as follows:

“Although individuals with known syndromic disorders were excluded from enrollment in the SVR and ISV trials, we found that 14% of the SV-CHD cohort harbored a damaging genetic variant in a gene known to cause syndromic CHD, including *KMT2D*, *NOTCH1* and *ANKRD11* as the most common recurring damaged genes. Similar to this cohort, the PCGC also reported a higher-than-expected number of participants with damaging genotypes in syndromic genes, despite a known genetic syndrome as an exclusion criterion.⁷ The inherent variable expressivity of cardiac syndromic disorders and/or potential challenges in identifying syndromic features in infancy may, in part, explain these findings. **Importantly, our estimates of the impacts of damaging genotypes in syndromic genes (or their absence) are underestimated by the initial exclusion of known syndromic patients. The power of this Bayesian approach is that we can tease apart the contribution of syndromic genes (or other gene classes) to clinical outcomes and thereby predict risk across all patients.**”

Results

Please comment on the 14-month survival rates of the two trials. Are these rates comparable to other studies and is there an explanation for the difference between the two cohorts (57% and 73%)? This may inform the reader about the generalizability of the study results.

Thank you for catching this typo. The rates mentioned were the percentage that survived AND had available neurodevelopmental scores. The survival rates are higher and have been previously published. These references have been added and we have edited the first paragraph of the Results to clarify the stepdown in numbers for each of the analytic cohorts and their basic demographics as noted above with Supplementary Table 1. The results now read:

“Of the original 555 SVR and 230 ISV trial enrollees, survival rates have been previously published.^{15,17} Of those that underwent WES, there were 21 individuals who were enrolled in both SVR and ISV trials. For this study, a total of 304 participants underwent WES, neurodevelopmental assessment and length measurement at 14 months of age and form the primary study cohort. Patient-parent trio WES was available for 86 participants. Additionally, the selected pre-

and post-operative clinical and demographic variables were available for a subset of 179 SVR participants. Distribution of enrollment demographics including sex, race, ethnicity, cardiac anatomy, gestational age, and birthweight were similar across the various analytic cohorts (Supplementary Table 1)."

The authors state that "Of the 555 and 230 individuals enrolled in the SVR and ISV trials, respectively, 57% and 73% survived to the 14-month endpoint and 36% and 67% underwent WES". Does the 36% and 67% refer to the 555 and 230 individuals? This is not clear.

The percentages described refer to the original cohort. This has been clarified with the changes noted in the response to the previous question and with Supplementary Table 1.

Can the authors provide any information why exome sequencing was not conducted in a large proportion of patients?

Due to the era and original study design, DNA collection was not required for enrollment. In order to clarify, we updated the Limitations section as follows:

"There are several limitations inherent to this study. For example, DNA collection was not a requirement for participation and thus, sample procurement was inconsistent across the cohort. Moreover, DNA samples were not obtained at enrollment, but rather at subsequent time points in these longitudinally followed cohorts, thereby introducing survival bias. Some DNA samples were collected at a few years of age which would have been well after the 14-month clinical endpoints included in these analyses. Thus, patients who died early may have harbored highly damaging genotypes that are not accounted for in this cohort of infants that survived at least until the 14-month endpoint..."

The authors state that the subset of patients with trio exome sequencing is $n = 108$. In line 246 it is stated that the subset of patients with trio exome sequencing is $n = 86$. Why are 22 patient trios missing in this analysis?

We thank the reviewer for pointing out this discrepancy. We have confirmed the correct number as 86 patient trios analyzed and the 108 was a typo. This has been corrected.

In the network analysis, MDI and PDI are included and interpreted as outcomes and risk factors. Since MDI and PDI are assessed at the same time, the authors cannot infer about a causal pathway and therefore these variables should not be interpreted as "risk factors" for the other outcome. I suggest to either calculate separate networks for each outcome measure (PDI, MDI, LAZ) or not using the term "risk factor" for outcome variables.

Using multi-variate regression analyses, one cannot simply swap x and y variables without creating a new model. One can however do this using a Bayesian network, indeed this is a well-documented and primary motivation for this application. This works because the Bayesian network models the complete joint distribution for the dataset – every possible combination of outcomes conditioned upon every possible combination of risk factors. This allows for one model to ask and answer every question; an advantage to this approach.

That being said, you are correct in that MDI and PDI are assessed at the same time, creating confusion as to the use of the term “risk factor”. Thus, we now describe our rationale for the term risk factor in the Methods section as follows:

“Outcomes included the Bayley Scales of Infant Development (BSID-II) and length measurement (length for age z-score, LAZ) for both the SVR and ISV trials at 14 months of age. BSID-II subdomain scores included the Psychomotor Developmental Index (PDI) and Mental Developmental Index (MDI), wherein the standardized mean scores are 100. While it is true that MDI and PDI are assessed at the same time, we consider these variables within the Bayesian Networks as both outcomes and risk factors, in order to highlight the ability of our approach to capture and quantify synergistic (non-additive) relationships between variables under study. An advantage of our approach is that the use of a Bayesian network allows us employ a single model to explore the impact of highly correlated variables singly or jointly, as outcomes and as risk factors to generate the results shown here.”

Since the authors use a data-driven machine learning/AI approach to investigate the conditional dependencies among genetic, demographic and clinical variables, the model should be validated in an independent dataset. Could the authors discuss this in the discussion section.

We agree that a validation cohort is important to support this preliminary work. Unfortunately there are very few CHD cohorts that are so well phenotyped and genotyped. PCGC and PHN investigators are currently working on creating additional cohorts for such validation. While that work is ongoing we thought it was important to get these data out. The key goal of our study was to apply and highlight the power of Bayesian Networks for risk prediction in pediatric cardiology. This is the first demonstration of this AI approach in this context. We have added to the Discussion the lack of a validation cohort in the Limitations section as follows:

“Finally, the goal of this study was to highlight the advantages of an AI-based Bayesian Networks approach as a proof-of-principle analysis, leveraging existing ISV and SVR clinical and genetic data. In this context, the risk predictions are directly relevant to the study cohort, but cannot be more widely applied without a validation cohort.”

Please provide confidence intervals in Figure 3 and 4 as well as when you report relative risks in the text.

When not visible in Figure 3 and 4, confidence intervals are within the symbols (now stated in the figure legend).

Discussion

Please comment on female sex being a significant risk factor for poorer neurodevelopmental outcome in the model. This is unusual as most previous studies identified male sex a risk factor for poorer neurodevelopmental outcome.

In this cohort and as an isolated variable, female sex only modestly increased the risk of a low MDI score 1.77-fold (while increasing risk of low PDI 0.61-fold).

However, in combination with a low PDI score, female sex further increased risk of low MDI 76-fold. We agree that this is not what has been previously described. This highlights the previously described limitations as it may be related to the selection and survival biases unique to this cohort. As noted above, additions have been made to the limitations section regarding the generalizability of the findings in this proof-of-principal study for this specific cohort.

Reviewer #2

The authors do not provide detailed information on the discretisation of many of the continuous variables.

We agree that the dichotomization of continuous variables can be debated. As noted in the responses to Reviewer 1, in most instances we used previously established cut-offs for risk, such as birthweight less than 2.5 kg and prematurity <37 weeks. We have added references to the Methods section to clarify the choices when grid search and ROC were not performed.

“The pre- and post-operative clinical and demographic variables in the smaller subset of 179 SVR participants were less amenable to optimization using this approach. Therefore, we used previously published cutoffs for dichotomization previously associated with outcomes in this population^{18,19} as follows: birthweight (< 2500 gram, low), ventilation following the Stage I palliation surgery (> 7 days, prolonged), gestational age (<37 weeks, preterm), and US census socioeconomic score (SES, low score < -0.3, Range -11.5 to 17.9).”

The use of AI-methodology resulted in finding a potentially damaging variant in 44% of the cohort. This seems very high- there has not been any genetic study in CHD (but also in most complex diseases) that has been able to identify such a large percentage. I think this approach includes variants (especially missense) variants that are most probably benign.

A quick exploration of these variants showed that several of them are not predicted pathogenic by multiple pathogenicity prediction algorithms (I used varsome). So, I argue that this approach includes variants that might not be contributing at all the to cases' phenotype. In my opinion, a stricter variant prioritisation is required. However, this would result in a much lower percentage of cases with potentially pathogenic variants and given the size of the cohort, might make the current analysis impossible.

Attributing disease causation to prioritized variants remains an inexact science, as discussed in our review of this topic (Eilbeck, Quinlan, Yandell. Settling the score: variant prioritization and Mendelian disease. *Nature Reviews*, 2017). In the current study, our motivation was not to define the genetic etiology of single ventricle CHD, but rather a proof of principle exploration of the contribution of (potential) damaging genetic variants to specific clinical outcomes. Our definition of a potentially damaging genetic variant and rationale for using the specified cut-off are described in the Methods section. A detailed description of the GEM AI tool and its benchmarking against clinically solved cases can be found in De La Vega, F.M. *et al.* Artificial intelligence enables comprehensive genome interpretation and nomination of candidate diagnoses for rare genetic diseases. *Genome Medicine* 13, 153, 2021. In that study we found for 30% of clinically diagnosed probands, one or both of the causative variants was not in

ClinVar (or Varsome) at time of diagnosis. Our current findings are consistent with this— and further highlight the utility GEM for discovery new disease-causing variants. This is especially important for association studies when the dataset of sequenced individuals is small, as it is here.

You are no doubt correct that this approach includes variants that might not be contributing to all the cases' phenotype. A stricter variant prioritization would certainly increase specificity, but at the cost of reduced sensitivity. Although false positives will dilute the genetic signals, truly benign variants are *a priori* not positively correlated with negative outcomes. Thus, the magnitudes and statistical significance of the associations between damaging genotypes and clinical variables that we report should be thought of as *lower bounds*.

We expand on these points in the Limitations paragraph as follows:

The definition of a “damaging” genetic variant is a topic of intense debate (PMID: 28804138). Our rationale for defining a damaging variant was based on a recent study benchmarking the AI tool against clinically solved cases²⁴. Stricter criteria for a damaging classification will allow for fewer false positives, but at the expense of false negatives and *vice versa*. However, the expectation is that false positives will dilute genetic signals, but truly benign variants are *a priori* not positively correlated with negative outcomes. Thus, the magnitudes and statistical significance of the associations between damaging genotypes and clinical variables reported here should be thought of as lower bounds.

The authors comment on the CHD cohort consisting of mainly non-syndromic CHD cases. However, they search for genetic variants in syndromic CHD genes. There is no exploration of matching of phenotypes between the cases and the genes. In Table 1 (and Sup Table 1), the diseases associated with the variants found are very specific – have there been any exploration of any phenotypic features in the corresponding cases that harbour these variants?

We agree that it is important to confirm whether the variants in syndromic genes were associated with identified syndromes. Given the era of the study and the variable follow-up, we cannot confidently explore matching cases and genes in all cases. Anecdotally, however, because the SVR trial had subsequent follow-up trials there were some instances where a subject who was not excluded from the original study for a syndrome was later noted to have a syndrome. We have included in the discussion an example of these correlations, although it is not an exhaustive search of this list. The Limitation paragraph now includes:

“The later DNA analysis also highlights the limitations of the original exclusion criteria, as some individuals in SVR were later found (during enrollment in SVR extension studies) to have syndromes, such as Kabuki Syndrome, consistent with our variant analysis.”

REVIEWERS' COMMENTS:

Reviewer #1 (Remarks to the Author):

The authors have done an excellent job addressing all comments.

I have only two comments: one refers to the abstract but that is, as you mentioned, an editorial decision (increasing the size of the abstract).

Survival rates: the authors mention that they now refer to the previously published survival rates. I suggest that the authors still mention the two survival rates (for each cohort).

There may be a typo in the sentence "An advantage of our approach is that the use of a Bayesian network allows us employ a single model to explore the impact of highly correlated variables singly or jointly, as outcomes and as risk factors to generate the results shown here." -allows us TO employ?

Reviewer #2 (Remarks to the Author):

The authors have addressed my comments and have clarified that their approach might include variants that do not contribute to the cases' phenotype, but truly benign variants will be a priori not correlated with negative outcomes. Ideally, a validation cohort is required but the authors have highlighted that this is a proof-of-principle study.

I have a couple of minor comments:

1. In the new Sup Table 1 the authors present a diverse ethnicity of the genetic samples – can population stratification seen in the genetic data affect this analysis?
2. The authors mention they used a threshold of 0.69 for the Bayes Factor- do they mean logBF?

RESPONSE TO REVIEWERS

We thank the Reviewers and Editors for their thoughtful and insightful comments and have incorporated their comments into the body of the manuscript. A summary of the reviewer comment and our point-by-point response are listed below.

Reviewer #1 (Remarks to the Author):

The authors have done an excellent job addressing all comments. I have only two comments: one refers to the abstract but that is, as you mentioned, an editorial decision (increasing the size of the abstract).

Seeing no additional comment from the editor, we have left the abstract at its current length.

Survival rates: the authors mention that they now refer to the previously published survival rates. I suggest that the authors still mention the two survival rates (for each cohort).

These rates have been added (69% and 86% for SVR and ISV, respectively).

There may be a typo in the sentence "An advantage of our approach is that the use of a Bayesian network allows us employ a single model to explore the impact of highly correlated variables singly or jointly, as outcomes and as risk factors to generate the results shown here." - allows us TO employ?

This is a typo and has been corrected as recommended.

Reviewer #2 (Remarks to the Author):

The authors have addressed my comments and have clarified that their approach might include variants that do not contribute to the cases' phenotype, but truly benign variants will be a priori not correlated with negative outcomes. Ideally, a validation cohort is required but the authors have highlighted that this is a proof-of-principle study.

I have a couple of minor comments:

1. In the new Sup Table 1 the authors present a diverse ethnicity of the genetic samples – can population stratification seen in the genetic data affect this analysis?

The potential impact of population stratification is minimized by the use of our AI tool, GEM, which is ancestry aware in that GEM evaluates each variant based on that ancestry's allele frequency in large-scale in gnomAD.

2. The authors mention they used a threshold of 0.69 for the Bayes Factor- do they mean logBF?

In the Methods, we define the GEM score as a Bayes factor (BF), i.e. the $\log_{10}(\text{ratio})$ of the posterior probabilities of two models. In this case, the numerator of the ratio is the posterior probability that the prioritized genotype damages the gene in which it resides and explains the proband's phenotype, whereas the denominator is the posterior probability of the contrapositive model, i.e. that the variant neither damages the gene nor explains the proband's phenotype. In keeping with standard best-practice

interpretation guidelines, a BF of 0.69 is usually taken as 'modest' support for model in the numerator.; see Kass and Raftery (1995). For more on GEM scores see De la Vega et al. Our rationale for using a GEM score ≥ 0.69 is further described in the Methods, with appropriate citations.

*Robert E. Kass & Adrian E. Raftery (1995). "Bayes Factors" (PDF). *Journal of the American Statistical Association*. 90 (430): 791. doi:10.2307/2291091. JSTOR 2291091.*